# Mating-Type Analysis in *Diaporthe* Isolates from Soybean in Central Europe

**DOI:** 10.3390/jof11040251

**Published:** 2025-03-25

**Authors:** Behnoush Hosseini, Lena Sophia Käfer, Tobias Immanuel Link

**Affiliations:** Department of Phytopathology, Institute of Phytomedicine, Faculty of Agricultural Sciences, University of Hohenheim, Otto-Sander-Str. 5, 70599 Stuttgart, Germany; lena.kaefer@uni-hohenheim.de

**Keywords:** mating-type genes, mating-type locus, *Diaporthe longicolla*, *Diaporthe caulivora*, phylogeny

## Abstract

Species of the genus *Diaporthe* have a mating-type system with the two mating types MAT1-1 and MAT1-2, like other ascomycetes. They can either be heterothallic, which means that any isolate only possesses one of the two mating types and needs a mating partner for sexual reproduction, or homothallic, which means that they possess both mating types and are self-fertile. For several *Diaporthe* species, no sexual reproduction has been observed so far. Using PCR with primers specific to the defining genes *MAT1-1-1* and *MAT1-2-1*, we determined the mating types of 33 isolates of *Diaporthe caulivora*, *D. eres*, *D. longicolla*, and *D. novem* from central Europe. In addition, we partially sequenced the mating-type genes of 25 isolates. We found that different *D. longicolla* isolates either possess *MAT1-1-1* or *MAT1-2-1,* making the species heterothallic, which is in contrast to previous studies and the general assumption that *D. longicolla* only reproduces asexually. *D. eres* and *D. novem* were also found to be heterothallic. Using genomic sequence information and re-sequencing of DNA and RNA, we identified the *MAT1-1-1* gene in *D. caulivora* and present here the full sequence of the mating-type locus of this homothallic species. Finally, we used sequence information from *MAT1-1-1* and *MAT1-2-1*, respectively, for improved phylogenetic resolution of our isolates.

## 1. Introduction

Fungi, like other eukaryotic organisms, have developed sexual reproduction to enable recombination of DNA (from genes to genomes) in populations. In addition to the standard male–female dichotomy, classes exist that have many different mating types, where specimens can mate with all other specimens of different mating types except for those that have the same mating type [1]. In addition, there are a number of parasexual phenomena that can also ensure recombination.

The ascomycetes that belong to the true fungi, however, rely on a simple bipolar system with only two mating types [1], recently denominated MAT1-1 and MAT1-2 [2,3,4]. In principle, the loci, or the genes making up the loci, that define the mating types are allelic, but because they are very different from each other, they have been defined as idiomorphs rather than alleles [1,2]. For almost all populations for which sexuality has been observed, successful mating producing fertile offspring is used as an overarching criterion to define the mating specimen as belonging to the same species. This way, mating can be used to define species limits. Methods for testing species limits based on molecular phylogenies like genealogical concordance, phylogenetic species recognition, and Poisson tree processes can only substitute for this, even though they are very useful when mating is difficult to observe [5]. Species barriers are often equivalent to mating barriers, and mating barriers can be defined in mating-type genes. Mating-type genes were observed to evolve rapidly but seem to be conserved within a species [6], so sequence analysis of mating-type genes can also contribute to a better definition of species limits in phylogenies [1].

For populations, including fungal populations, it makes a difference whether recombination through sexual reproduction happens. Therefore, it is interesting to study the presence of different mating types in a population. Also, specimens with different mating types might differ somewhat in their physiology, perhaps even in their virulence against different host species.

The genus *Diaporthe* has the typical mating-type system for the Ascomycota [3]. The mating types are mainly defined by the presence of either one of the genes *MAT1-1-1* for MAT1-1 or *MAT1-2-1* for MAT1-2 [2]. These genes are coding for regulatory proteins that induce differentiation of generative tissues as well as the mating factor pheromones [7]. Therefore, the mating types of different specimens, or rather strains or isolates, can be determined through testing for the presence of either of these two genes. Since the genes are rather different, this can be achieved by PCR with primers specific either for *MAT1-1-1* or for *MAT1-2-1*. For this purpose, Santos et al. [4] designed primers that can amplify either part of *MAT1-1-1* or *MAT1-2-1* across *Diaporthe* spp.

Because *Diaporthe* spp. show considerable variability in their morphological features, it is very difficult to use morphology for species identification and even more so for species definitions [8,9]. Because of these difficulties, association of *Diaporthe* strains to different plant species has been used for species definitions, which led to a strong overestimation of *Diaporthe* species numbers, since several *Diaporthe* species can infect more than one plant species, and some can infect many [10]. On the other hand, plants can be infected by different *Diaporthe* species; for example, soybean can be infected by *D. aspalathi*, *D. caulivora*, *D. eres*, *D. longicolla*, *D. novem*, and *D. sojae*, and probably even more *Diaporthe* spp. [11,12,13]. Recently, molecular phylogeny has been used to define the clades in the genus, which led to a much clearer idea of species number and host associations in the genus [5,14]. However, a lot of work still has to be carried out before the genus is fully resolved and the present study also hopes to contribute to this.

In a previous study our group showed the presence of the four species *Diaporthe caulivora*, *D. eres*, *D. longicolla*, and *D. novem* on soybean in central Europe [9]. Hosseini et al. [9] gained 32 isolates of the different species and performed phylogenetic analysis with these isolates. Here, we determined the mating types of these isolates and we used sequence information on the *MAT1-1-1* and *MAT1-2-1* genes to further refine phylogenetic information for our isolates. Despite the indirect role of these genes in defining the mating type, we assumed that sequence comparisons on these genes could further contribute to defining the species limits.

*D. caulivora* is a homothallic species that readily forms perithecia and ascospores in vitro and on its host plants; it preferentially reproduces sexually, whereas asexual conidia are seldom found [9]. So far, only the *MAT1-2-1* gene was found for *D. caulivora* [15], which somewhat contradicts self-fertility, for which normally both mating-type genes should be present.

*D. eres* is considered a species complex [14], and we also found sequence variation in ITS, *TEF,* and *TUB* genes between our isolates that were classified as *D. eres* [9]. We anticipated additional information on whether our isolates do belong to the same species or whether they constitute different species by sequence comparisons and mating experiments. Therefore, we needed to first determine the mating type of our isolates before setting up mating experiments with isolates of opposing mating type.

Among the *Diaporthe* species, *D. longicolla* is probably the most important pathogen for soybean as it is the main causal agent of Phomopsis seed decay and is also associated with stem blight [9,16,17]. For *D. longicolla,* only asexual reproduction through conidia has been observed, so far. Because of this, the species was known as *Phomopsis longicolla*, the name for the anamorph, for a long time. Only recently it was renamed *D. longicolla*, according to the rule that there should be only one name for a species or a genus, respectively. In Hosseini et al. [9], all *D. longicolla* isolates were 100% identical, and they also showed full identity to the type strains for *D. longicolla*, which indicates strong genetic homogeneity within the species. This homogeneity could be due to the lack of sexual recombination and more or less clonal reproduction. Theoretically, a genetic reason why sexual reproduction does not occur could be that only one mating type is present in the population. This famously happened to *Phytophthora infestans* in Europe. Contrary to this expectation, an isolate tested for its mating type in a previous study [15] showed the presence of both *MAT1-1-1* and *MAT1-2-1,* indicating that *D. longicolla* should be homothallic and self-fertile. Here, we expected to expand on these results with more isolates.

*D. novem* has only recently gained species status [15]. It was found to be heterothallic. Our isolates, which clearly clustered with the *D. novem* type strains, did show sequence variation, however. Hence, similar to the situation in *D. eres,* successful mating between *D. novem* isolates would be very helpful in determining the species limits for *D. novem*.

The original aim of this study was to find further genetic differences between our *Diaporthe* isolates beyond what we already found in ITS, *TEF,* and *TUB* genes. These genetic differences could then be correlated to any morphological, physiological, or virulence differences between the isolates. What we describe here are mainly novel findings considering the mating-type distribution in *D. longicolla* and the mating-type locus in *D. caulivora*. In addition, we present phylogenies of our isolates based on or amended with sequences of the mating-type genes and the results of our mating experiments.

## 2. Materials and Methods

### 2.1. Fungus Material

All *Diaporthe* isolates were from our own (Department of Phytopathology, University of Hohenheim) lab collection. Collection, isolation, and culture of the isolates, as well as descriptions and phylogeny of most isolates, are delineated in [9]. Two additional *D. caulivora* isolates (DPC_HOH33 and DPC_HOH34) and one *D. eres* isolate (DPC_HOH35) that were isolated from a field near Tübingen in 2022 were also included in this study. Sequences identifying the isolates were submitted to NCBI: ITS accession numbers OR767918 for DPC_HOH_ITS33, OR767919 for DPC_HOH_ITS34, and OR783466 for DPC_HOH_ITS35; *TUB* accession numbers OR785372 for DPC_HOH_TUB33, OR785373 for DPC_HOH_TUB34, and OR785374 for DPC_HOH_TUB35; and *TEF* accession numbers OR785377 for DPC_HOH_TEF33, OR785378 for DPC_HOH_TEF34, and OR785379 for DPC_HOH_TEF35.

### 2.2. Mating Experiments

Isolates of opposing mating types from the heterothallic species were mated to check for correct species definition and to observe teleomorph structures of these species.

We used the method described by Brayford [18] with some modifications. Plates with 2% water agar or with synthetic medium according to [19] onto which autoclaved soybean stems had been placed and alternatively also water agar plates or plates with 2% agar and 1:4 diluted PDA with autoclaved wild fennel branches or pine needles on them were used for the mating experiments. On opposing ends of these plates were placed agar plugs with the *Diaporthe* sp. isolates that had previously been grown either on PDA or 1:5 diluted PDA. The plates were incubated at room temperature with different lighting conditions: continuous darkness, continuous white light, 12 h:12 h light/dark cycle, continuous UV light, and three days of darkness followed by 12 h:12 h light/dark cycle. The incubations were kept up for three months with regular checking of the plates for perithecia.

Observation of perithecia was regarded as proof of successful mating. No observation of perithecia was recorded as a negative result. All matings with all conditions mentioned above were performed in three replicates. As a control, all isolates were also paired with themselves.

### 2.3. PCR Analysis

#### 2.3.1. DNA and RNA Preparation

Mycelia of ten-day-old *Diaporthe* cultures were used for nucleic acid preparations. Roughly 100 mg were scraped from the APDA plates and homogenized by 20 s vortexing with microbeads (Lysing Matrix E tubes, Fast Prep-24™, MP Biomedicals GmbH, Eschwege, Germany) in the corresponding lysis buffer.

DNA was prepared using the method described by Liu et al. [20]. The lysis buffer (500 µL per tube or preparation) was 400 mM Tris-HCl (pH 8.0), 60 mM EDTA (pH 8.0), 150 mM NaCl, and 1% sodium dodecyl sulfate. After homogenization and lysis, 150 µL of 3 M KAc (pH 4.8) was added, and the tubes were centrifuged at 10,000 rcf for 1 min. The supernatant was transferred to a fresh tube, an equal volume of isopropanol was added and the tubes were centrifuged at 10,000 rcf for 2 min. The DNA pellet was washed with 300 µL 70% ethanol, spun again at 10,000 rcf for 1 min. The supernatant was removed again and the pellet was left to air dry. Then, it was dissolved in 50 µL distilled water.

RNA was prepared using the RNA Plant Mini Kit (Qiagen, Hilden, Germany) following instructions in the manufacturer’s manual. To remove residual genomic DNA, RNA was treated with 1 U DNaseI (Thermo Fisher Scientific, Waltham, MA, USA) per 1 µg RNA in a 10 µL reaction with 30 min incubation. Reverse transcription of the RNA to cDNA was performed using the Tetro™ cDNA Synthesis Kit (Meridian Bioscience Inc., Cincinnati, OH, USA) with random primers. No reverse transcriptase controls were generated for all RNA samples.

Nucleic acid concentrations were measured spectroscopically using the OD_260_ method.

#### 2.3.2. PCR Conditions for Detecting the Presence of *MAT1-1-1* and *MAT1-2-1*

For PCR detection of the mating-type genes, PCR was run using Taq DNA polymerase (Thermo Fisher Scientific, Waltham, MA, USA) with the standard buffer and added MgCl_2_ in a 25 µL format (ca. 50 ng DNA template, 12.5 pmol of each primer, 5 nmol dNTP mix, 1 U Taq, 5 µL 5× buffer). We used the same cycling conditions as [4]: 5 min 95 °C, then 40 cycles with 30 s 94 °C, 30 s 50 °C (for MAT1-1-1FW and MAT1-1-1RV) or 56 °C (for MAT1-2-1FW and MAT1-2-1RV), 1 min 72 °C, and final elongation for 10 min 72 °C.

For detection of the mating-type genes in *D. longicolla* using primers DLmat111_fs and DLmat111_rs or DLmat121_fs and DLmat121_rs, both the mixes and the cycling conditions were the same as above except for annealing temperature, which was raised to 60 °C to correspond to the predicted melting temperatures of the new primers.

For primer sequences see Table 1.

#### 2.3.3. PCR Conditions for Producing a Template for Sequencing

All amplifications where it was intended to use the PCR product for sequencing were performed using Phusion DNA polymerase (Thermo Fisher Scientific, Waltham, MA, USA) in 40 µL reactions with Phusion HF buffer. Mixes were prepared as proposed by the manufacturer. Cycling conditions varied: Initial denaturation always was 98 °C for 30 s; also, the denaturation step was 98 °C for 10 s. But depending on the primers, annealing temperatures varied from 55 °C to 65 °C, and sometimes we also employed gradient protocols to ensure amplification. Elongation at 72 °C varied between 15 s (sequencing MAT1-1-1 and MAT1-2-1 fragments for the phylogenies), 30 s, and 45 s (sequencing the MAT1/2 locus of *D. caulivora*, smaller and larger (nested PCR) fragments). Denaturation, annealing, and elongation were repeated for 35 cycles. Final elongation was 72 °C for 10 min. Since the mating-type genes are not highly expressed, it was necessary for RT-PCR in several cases to use nested PCR to receive sufficient PCR product for sequencing. In these cases, the PCR product from the first round PCR was diluted by 1:100 and used as a template for the second round PCR. All primers used for re-sequencing the MAT locus in *D. caulivora* and in *D. longicolla* are listed in Appendix A.

### 2.4. Cloning

When we did not succeed in obtaining clear bands or pure PCR products adequate for use in Sanger sequencing, these PCR products were cloned. For this we used the Clone Jet PCR Cloning Kit (Thermo Fisher Scientific, Waltham, MA, USA). We followed the instructions given in the product information manual, using the blunt-end cloning protocol with 1 µL non-purified PCR product. The ligation reactions were incubated at RT for 30 min. The ligation mixes were transformed into *E. coli* DH5α using chemically competent cells made in our own lab following protocol 1.24 in [21]. Colony PCR (direct PCR on colonies of DH5α transformants) to confirm the presence of the correct inserts was performed using Taq DNA polymerase (Thermo Fisher Scientific, Waltham, MA, USA) with the standard buffer and added MgCl_2_ in a 10 µL format. Primers specific for the expected inserts were used and annealing temperatures were changed according to the primers and elongation times according to the expected amplicon length. 5 mL overnight cultures in LB with ampicillin were set up for transformants with the correct insert and from these plasmid preparations were performed using the peqGOLD Plasmid Miniprep Kit I (VWR International, Darmstadt, Germany).

### 2.5. Sequence Analysis

Sanger sequencing was performed at Microsynth Seqlab GmbH, Göttingen, Germany, using sequence-specific primers both when PCR products or plasmids were sent for sequencing.

Sequence assemblies and editing were performed using SeqMan™II and EditSeq™ from Lasergene ver. 5.07 (DNASTAR, Madison, WI, USA). Multiple sequence alignments (of more dissimilar sequences) were performed using ClustalW as implemented in BioEdit (version 7.1.3.0, [22]). Phylogenetic trees were constructed with the maximum composite likelihood method [23] using MEGAX (version 10.0.5) [24]. The evolutionary history was inferred by using the Maximum Likelihood method and the Tamura–Nei model [25]. Initial tree(s) for the heuristic search were obtained automatically by applying the Maximum Parsimony method. Codon positions included were 1st + 2nd + 3rd + Noncoding. All positions with less than 80% site coverage were eliminated, i.e., fewer than 20% alignment gaps, missing data, and ambiguous bases were allowed at any position (partial deletion option). Open reading frames were predicted using GeneRunner (version 6.5.52 beta). Graphic representation of the gene locus was constructed using Incscape (version 1.3.2).

## 3. Results

### 3.1. Mating-Type Analysis Shows That D. longicolla, D. novem, and D. eres Are Heterothallic Species

After producing 32 isolates from *Diaporthe* spp. corresponding to *D. caulivora*, *D. eres*, *D. longicolla*, and *D. novem* and phylogenetic analysis of the isolates [9], we set out to further differentiate between those strains that were identical in all three genes (*TUB*, *TEF*, ITS) used for the phylogeny. One way to differentiate the strains was to check for different mating types. This was performed using the PCR method introduced by Santos et al. [4]. 30 of the isolates described in [9] and the three additional isolates listed in Section 2.1 were included in this study.

For *D. longicolla*, we had successful amplification with the primers for *MAT1-1-1* in seven isolates and with the primers for *MAT1-2-1* in 14 isolates (Figure 1a). None of the isolates had both mating types. Since for *D. longicolla* (also *Phomopsis longicolla*) only asexual reproduction has been observed so far, this finding that established *D. longicolla* as a heterothallic species was surprising. Also, a previous study by Santos et al. [15] found both mating-type genes in their *D. longicolla* isolate. Because of this, we repeated the experiment with the same result.

To be entirely sure, we also designed primers specific to the *D. longicolla MAT1-1-1* and *MAT1-2-1* genes. For this, we first searched the *D. longicolla* genome sequences published by Li et al. [17] for the two genes and found *MAT1-1-1* in the assembly corresponding to *D. longicolla* strain TWH P74 and *MAT1-2-1* in the assembly corresponding to *D. longicolla* strain MSPL 10-6 78069. We resequenced parts of *MAT1-1-1* and *MAT1-2-1* in our isolates and found the sequences identical to the published sequences. Based on this, the primers DLmat111_fs and DLmat111_rs and DLmat121_fs and DLmat121_rs were designed (Table 1). The PCR using these species-specific mating-type primers fully confirmed the finding using the genus-specific primers (Appendix A). After our observation, it should be only a matter of time until the sexual structures of *D. longicolla* will be found.

For *D. eres,* two of our isolates had *MAT1-1-1* and four *MAT1-2-1*; none had both mating types (Figure 1b), corroborating *D. eres* as a heterothallic species.

For *D. caulivora* we could only amplify *MAT1-2-1* in all three isolates. Since Santos et al. [15] already speculated that their primers just cannot amplify the *D. caulivora MAT1-1-1* gene, we assumed that this is also the case for our isolates and then set out to find the *MAT1-1-1* gene (see Section 3.2).

In the three isolates representing *D. novem,* we could amplify *MAT1-1-1* twice and *MAT1-2-1* once, also in different isolates. This also corroborated *D. novem* as heterothallic.

### 3.2. Detection of MAT1-2-1 and MAT1-1-1 in D. caulivora and Sequence Analysis of the Full MAT1/2 Mating Locus of This Homothallic Self-Fertile Species

Using the primers designed by Santos et al. [4], we were able to only detect *MAT1-2-1* in the *D. caulivora* isolates DPC_HOH2, DPC_HOH33, and DPC_HOH34. This was expected, as the same result had also been achieved by this group. That the primers for *MAT1-1-1* could not bind was attributed to insufficient specificity of the primers. However, we found this finding not entirely satisfying; with self-fertile species, it is expected that they have both mating-type genes [2]. Hence, we decided to also search for *MAT1-1-1*.

For this search, it was quite advantageous that the genome sequence of *D. caulivora* was published recently [26]. Using BLAST (blastn https://blast.ncbi.nlm.nih.gov/Blast.cgi?PAGE_TYPE=BlastSearch&PROG_DEF=blastn&BLAST_SPEC=GDH_GCA_023703485.1 and tblastn https://blast.ncbi.nlm.nih.gov/Blast.cgi?PROGRAM=tblastn&PAGE_TYPE=BlastSearch&BLAST_SPEC=GDH_GCA_023703485.1&LINK_LOC=blasttab&LAST_PAGE=blastn, accessed on 7 July 2022) of the sequences published by [3] on this assembly on NCBI and also locally (BLAST 2.0) after downloading contig 6 that contains the MAT locus, we were able to find *MAT1-1-1* close to the other mating-type genes. The genes in the locus are ordered *MAT1-2-1*, *MAT1-1-2*, *MAT1-1-3*, and *MAT1-1-1*. Therefore, we could confirm that *D. caulivora* does indeed have both mating-type genes, as was expected.

To further confirm our finding and explore sequence variability within the locus, we re-sequenced the whole *MAT1/2* locus in our isolate DPC_HOH2, including the DNA lyase gene that is closely located and also used as an orientation by [3] in their sequencing efforts. All primers for sequencing the locus were designed using the sequence by [26].

The sequence stretch that we sequenced is 13,516 bp long. The sequence was deposited in GenBank with accession No. PP003215. Comparing the sequence from our DPC_HOH2 with that of D57 [26], we found the sequences to be mostly identical with only a few differences. In positions 808 and 809, DPC_HOH2 has CG, whereas D57 has TC; position 3150 in DPC_HOH2 is A instead of G in D57. From 3765 to 3777 is a stretch of Cs; here DPC_HOH2 has two less than D57. In position 5965 there is a T in DPC-HOH2 instead of an A in D57; in 6074 there is a G instead of an A. After position 11,038, the D57 has 60 bp more than DPC_HOH2 (the exact position of the extra sequence is not definite; 11,038 is the most 5′ position, and the most 3′ position is 11,143; the D57 sequence is repeating itself here). From 11,263 to 11,274 is a stretch of As, which is longer by one A in DPC_HOH2; in 12,822 there is a T instead of a C. Here, the base changes can be interpreted as definite SNPs, whereas the apparent insertions or deletions could also be PCR, sequencing, or assembly (of the D57 sequence) artifacts.

Another intriguing sequence feature came to our attention during primer design for re-sequencing—one sequence stretch of 451 bp occurs twice in the locus. The sequence is once located from 13,027 to 13,477, which means that it starts in the second intron of *MAT1-1-1* and spans the third exon and 268 extra bases downstream of the gene. The same sequence is also found in the reverse complement 6953–6503, which means that it is located directly downstream of *MAT1-2-1* in the reverse complement. Since *MAT1-2-1* is the only gene with reverse orientation in the locus, it is tempting to speculate that this insertion was connected to the event when *D. caulivora* transitioned to homothallism. To check whether this sequence was also inserted anywhere else in the genome, the D57 genome was BLASTed again with that sequence, and it was found that it occurs exactly twice at those two positions in the genome that we identified. The sequence also has no obvious transposon features, which indicated that this doubling of part of the MAT locus was a singular event.

To produce a definite exon–intron structure of the genes in the locus, we also did cDNA sequencing for all the genes. These results, together with the location of the five genes in the locus, are shown in Figure 2.

Once all the CDSs were known, we also checked the protein sequences for differences between the two isolates. For *MAT1-1-1,* no difference was found in the protein sequence; the SNP at position 12,822 is a synonymous exchange. For the other genes, the CDS in the DPC_HOH2 sequence was identical to the corresponding sequence in D57.

### 3.3. Use of Partial Sequences of MAT1-1-1 and MAT1-2-1 for Further Refinement of Phylogenies of D. caulivora, D. eres, D. longicolla, and D. novem

#### 3.3.1. Isolates and Phylogeny Based on ITS, *TEF*, and *TUB*

As detailed in Section 2.1, this study was conducted using our *Diaporthe* strains described in [9]. From additional isolates acquired in the meantime, we included three in this study and named them DPC_HOH33, DPC_HOH34, and DPC_HOH35. To optimally show the classification of these strains, we integrated them into the phylogeny containing our other isolates. Alignments of the ITS, *TEF,* and *TUB* sequences were performed and these were concatenated to a three-gene alignment and this way a three-gene phylogeny containing the new strains was built. The phylogeny is shown in Appendix A. Of the two new *D. caulivora* isolates, strain DPC_HOH33 showed 100% identity to DPC_HOH2, whereas DPC_HOH34 was identical to DPC_HOH4. The two pairs of isolates differ in one SNP in the ITS sequence. The new *D. eres* isolate sequence-wise was new to our collection with differences from all our previous isolates. These differences were found in the ITS and the *TEF* sequences, while in the *TUB* gene all isolates have the same sequence. Here, the new isolate once more adds to the sequence variability in *D. eres*.

#### 3.3.2. Phylogeny Based on *MAT1-1-1* and Combined *MAT1-1-1*, ITS, *TEF,* and *TUB* Phylogeny

After PCR using either the primers by Santos et al. [4] or alternatively species-specific primers for the gene, we acquired partial sequences for *MAT1-1-1* for most of our isolates. The sequences were submitted to NCBI GenBank. The accession numbers are listed in Table 2.

These sequences were aligned and a phylogeny was built on this alignment, which is shown in Figure 3.

The *MAT1-1-1* sequences show clear differences between the species but are identical within the species, which might indicate that there is little intraspecific variation in these genes. To allow direct comparison to what can be learned from the other genes used for phylogenies, an analysis was also performed in which *MAT1-1-1* was combined with ITS, *TEF*, and *TUB* (Figure 4).

Strikingly, the topologies of the trees are identical. Also, there seems to be very little intraspecific variation between the isolates with MAT1-1. With *D. eres,* it is conspicuous that two isolates that show identity in ITS, *TUB*, and *TEF* sequences also have the same mating type. Given the total number of *D. eres* isolates included in this study, this may be a coincidence, of course.

#### 3.3.3. Phylogeny Based on *MAT1-2-1* and Combined *MAT1-2-1*, ITS, *TEF*, and *TUB* Phylogeny

Similar to what was described in Section 3.3.2, we also acquired partial sequences for *MAT1-2-1* for most of our isolates. The accession numbers are also listed in Table 2. Also, as described in Section 3.3.2 for *MAT1-1-1,* a phylogeny was constructed for the isolates containing *MAT1-2-1* as shown in Figure 5.

Like *MAT1-1-1*, *MAT1-2-1* allows clear differentiation between the different species. The isolates belonging to each species cluster nicely together. However, for *D. eres* there exists sequence variation, which again illustrates the status of *D. eres* as a species complex.

Again, the topology of the combined tree (Figure 6) is very similar to the topology of the *MAT1-2-1* tree. Except for the branch depicting *D. eres*, here the isolates DPC_HOH3 and DPC_HOH7 that are identical in the *MAT1-2-1* sequence do not cluster directly but show up on different branches. According to our previous analysis [9], the sequence difference between DPC_HOH7 and DPC_HOH3 is mainly in ITS, with only a few different bases in *TEF*, whereas in *TUB* the sequences are identical, as in *MAT1-2-1*. Altogether, the fact that the *D. eres* isolates cluster together in all four studied genes strongly indicates that they are closely related; unfortunately, our analysis does not yet resolve the issue of whether they all belong to the same species, i.e., whether *D. eres* should be considered a single species or not.

### 3.4. Mating Experiments

Since we could identify isolates of different mating types in *D. longicolla* and for the first time identified the species as heterothallic, we assumed that it should also be possible to mate appropriate isolates of opposing mating type to observe perithecia with asci and ascospores of this species. Accordingly, mating experiments were performed using several different conditions as described in Section 2.2. In addition to the goal of observing the sexual structures of this species, isolates that successfully mate also belong to the same species, so with this experiment we also aimed to corroborate the phylogenies we inferred from sequence information and included isolates from *D. eres* and *D. novem*. Table 3 shows all isolates included in the experiment.

Unfortunately, no sexual structures could be recorded from any of the matings. Only once, in an early experiment, perithecia were seen in a plate on which DPC_HOH17 and DPC_HOH29 were paired. This was a water agar plate with a soybean stem and it was incubated in the dark. Because the significance of the observation had not yet been understood during this phase of the studies, no pictures were taken, so that, as already mentioned, no records exist of any teleomorph structures based on our isolates.

In parallel incubations, *D. caulivora* readily formed perithecia, indicating that the conditions used by us are overall conducive for sexual reproduction. That *D. longicolla* did not form perithecia may be attributed to a general reluctance of this species to mate. To identify the sexual structures of this species, it will be necessary to repeat mating experiments until conditions are identified that induce *D. longicolla* to mate.

The fact that no sexual structures could be observed for *D. eres* and *D. novem* could theoretically have different reasons. Either here too we just were not successful in finding the correct trigger for the mating, or our isolates that showed sequence variation in ITS, *TEF*, and/or *TUB* actually do not constitute a single species and a reproductive barrier exists between them.

## 4. Discussion

In this study, we have identified the mating types of 33 *Diaporthe* isolates from central Europe. In the course of these experiments we found that *D. longicolla* is heterothallic and provide evidence for the same mating system in *D. eres* and *D. novem*. In addition, we clarified the structure of the *MAT1/2* mating locus of *D. caulivora*, providing the basis for homothallism in this species.

### 4.1. D. longicolla Is Heterothallic, but Matings of Isolates with Different Mating Types Did Not Produce Sexual Structures

Santos et al. [15] were the first to test an isolate of *D. longicolla* for its mating type. They found both mating types in the same isolate and interpreted this as an indication of purely anamorphic reproduction in *D. longicolla*. Here, we tested 21 *D. longicolla* isolates with two independent primer combinations and consistently detected either *MAT1-1-1* or *MAT1-2-1*, which was contradictory to what [15] found. Based on the number of unambiguous PCR reactions and the additional evidence found in the genome assemblies by Li et al. [17], we conclude that our findings must be correct. How [15] could have diagnosed both mating-type genes in their isolate is unclear. Feasible explanations are that either the isolate that they studied was not a pure isolate and contained a strain with the MAT1-1 mating type and another strain with the MAT1-2 mating type or that a simple PCR artifact happened.

Whether a species reproduces sexually or is asexual may not necessarily be defined by the presence or absence of mating-type genes, since there are apparently asexual species that do have mating-type genes, but is probably determined by other factors [1,7]. It is conceivable that epigenetic regulation or genes downstream in the mating signaling cascade hinder mating. Santos et al. [4] also describe *D. viticola* as a species that should be heterothallic based on the fact that they could find either the *MAT1-1-1* gene or the *MAT1-2-1* gene in different isolates, but that does not produce sexual structures. In addition to this, as described in Section 2.2, we have tested several different conditions for mating and waited for a long time for sexual structures to appear. Apparently, *D. longicolla* is very reluctant to engage in sexual reproduction. Nevertheless, we cannot find a satisfactory explanation for how a population that only reproduces asexually can maintain both mating-type genes in different isolates. Theoretically, in such a population, one of the mating-type genes should be lost over time. Therefore, it still seems likely that the correct conditions for teleomorph induction in *D. longicolla* just have not been found so far. While we have emphasized nutrient depletion in our matings, elsewhere important nutrient requirements like enough sugars, amino acids, or calcium are described [7]. Additional environmental triggers that we have not tested so far could be reactive oxygen species or pheromones. Consequently, we assume that the teleomorph of *D. longicolla* may still be found in future mating experiments. Unfortunately, finding and identifying *D. longicolla* perithecia from soybean in the field is very difficult and labor-intensive.

Another aspect that is puzzling in this context is the fact that, apart from the different mating-type genes, all our *D. longicolla* isolates and the type species are genetically identical. This again would indicate a clonal population structure based on asexual reproduction. Could it be that both mating-type genes were maintained in the population because asexuality in *D. longicolla* started only very recently?

### 4.2. Structure of the MAT1/2 Locus in D. caulivora

Self-fertility requires the presence of both mating-type idiomorphs in a single organism or the same nucleus. This can mean that two full loci are present or that only part of a locus fuses with another locus to result in a new mating-type locus that enables self-fertility [1]. Ref. [1] also argues that such a combined MAT1/2 locus could arise through an unusual recombination event integrating the additional gene into the locus. The *D. caulivora* MAT1/2 locus belongs to this latter category as it contains only one copy each of the genes *MAT1-1-2* and *MAT1-1-3*.

The mating-type loci described in [3] all show opposite orientation of *MAT1-1-2* and *MAT1-1-3* relative to the DNA lyase gene, and in the MAT1-2 locus, *MAT1-2-1* also has a different orientation than in the MAT1/2 locus of *D. caulivora*. Compared to this, the MAT1/2 locus of *D. caulivora* appears as if a MAT1-2 locus was inverted and then *MAT1-1-1* was added at the new 3′-end. Considering that we found a partial *MAT1-1-1* sequence in opposite orientation right next to the position of *MAT1-2-1*, one could also assume that the heterothallic ancestor of *D. caulivora* had a MAT1-1 locus with opposite orientation to the MAT1-1 loci described in [3]. Then, in a major recombination event (or maybe two events), *MAT1-1-1* was doubled, putting the additional copy downstream of *MAT1-1-3*, where we can find it now, and *MAT1-2-1* was inserted into the original *MAT1-1-1* site.

This is a highly speculative assumption that cannot be proven for *D. caulivora* but might be tested in a model fungus.

### 4.3. Implications for Species Definitions

Classically, specimens can be defined as belonging to the same species when they can mate with each other and have fertile offspring. Unfortunately, our mating experiments were almost all unsuccessful, even though we tested several conditions for mating and waited a long time for perithecia to emerge. Why the matings were unsuccessful cannot clearly be explained. For *D. longicolla*, where all tested isolates were 100% identical in their ITS, *TUB*, and *TEF* sequences, so that molecular evidence strongly supports that these isolates belong to the same species, there seems to exist a general reluctance of the species to mate, despite the existence of heterothallism as discussed in Section 4.1.

For *D. novem* and for *D. eres*, our isolates showed differences in the ITS, *TUB*, and *TEF* sequences. Nevertheless, the branches in the phylogeny defining the different species clades are strongly supported. Coincidentally, our isolates, which had different mating types, were all also different in the ITS, *TUB*, and *TEF* combination. Therefore, the lack of successful matings might indicate that our isolates that we classified as *D. novem* and *D. eres* constitute more than just these two species. For *D. eres*, which has long been considered to be a species complex [14], this could be a plausible interpretation. On the other hand, Hilario et al. [27] showed incongruences between phylogenetic trees of the *D. eres* species complex based on different single gene alignments and found molecular evidence for recombination between the species in the complex, which made them conclude that the whole *D. eres* species complex is, in fact, only one species. More research is necessary to clarify these open questions; probably many additional isolates will have to be included in mating studies to define the exact species boundaries.

## Figures and Tables

**Figure 1 jof-11-00251-f001:**
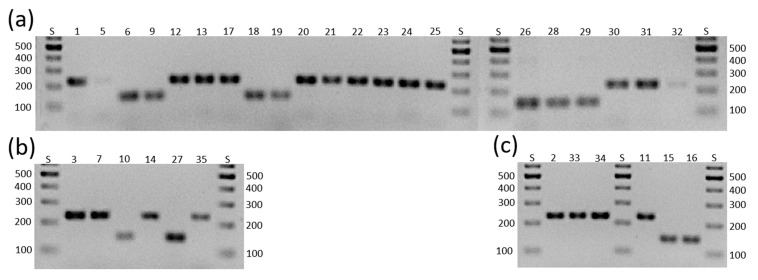
Mating-type analysis of 33 *Diaporthe* spp. isolates. (**a**) *D. longicolla*; (**b**) *D. eres*; (**c**) *D. caulivora* (2, 33, 34); and *D. novem* (11, 15, 16). S: GeneRuler 100 bp DNA ladder, band sizes given besides the gel pictures. Numbers above the lanes indicate isolates (DPC_HOHn). PCR products for *MAT1-1-1* are around 140 bp in size, and *MAT1-2-1* is 230 bp. PCRs using either the primer pair for *MAT1-1-1* or *MAT1-2-1* were run separately, but for the presented gels both PCR products of reactions run on the indicated isolates were mixed before adding them to the gel.

**Figure 2 jof-11-00251-f002:**
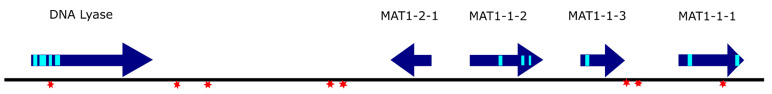
The *MAT1/2* mating-type locus of *D. caulivora* (DPC_HOH2). The black bar represents the 13,516 bp that were sequenced. Red asterisks indicate the positions of SNPs; blue arrows represent ORFs; lighter shading indicates introns; the headers indicate the different genes. Drawn to scale.

**Figure 3 jof-11-00251-f003:**
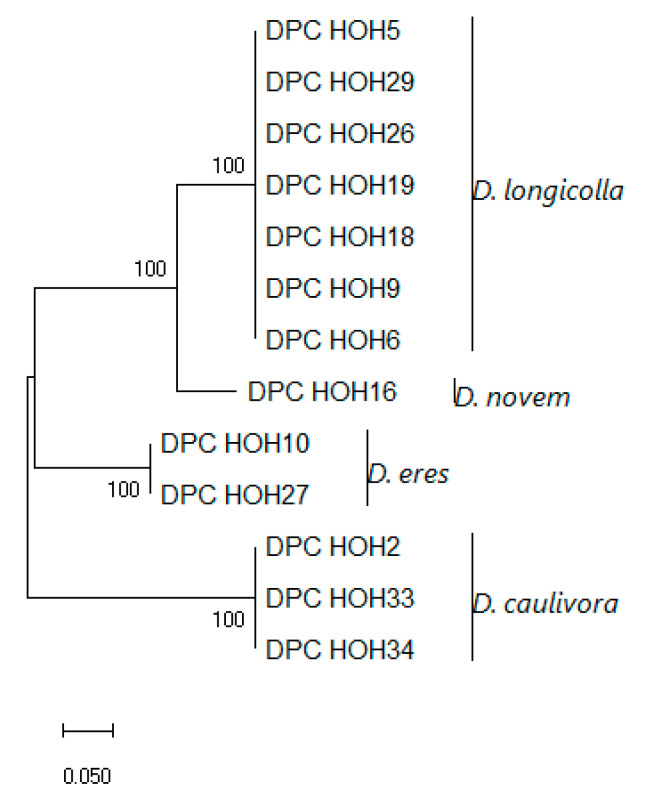
Evolutionary analysis of DPC_HOH isolates by the Maximum Likelihood method based on *MAT1-1-1* sequences. The tree with the highest log likelihood (−1740.01) is shown. The percentage of trees in which the associated taxa clustered together is shown next to the branches. The tree is drawn to scale, with branch lengths measured in the number of substitutions per site. This analysis involved 13 nucleotide sequences. There were a total of 499 positions in the final dataset.

**Figure 4 jof-11-00251-f004:**
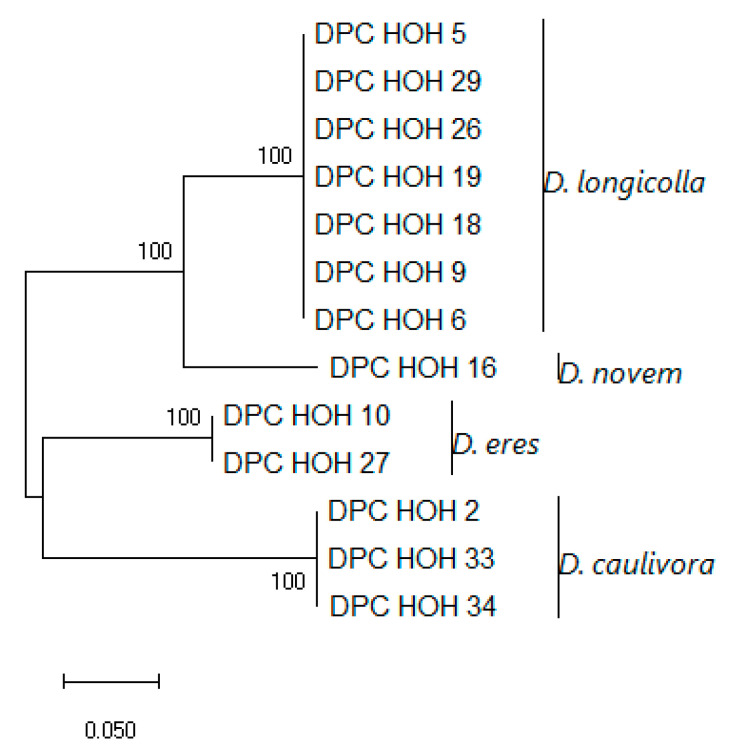
Evolutionary analysis of MAT1-1 DPC_HOH isolates by the Maximum Likelihood method based on ITS, *TEF*, *TUB*, and *MAT1-1-1* sequences. The tree with the highest log likelihood (−5245.28) is shown. The percentage of trees in which the associated taxa clustered together is shown next to the branches. The tree is drawn to scale, with branch lengths measured in the number of substitutions per site. This analysis involved 13 nucleotide sequences. There were a total of 1703 positions in the final dataset.

**Figure 5 jof-11-00251-f005:**
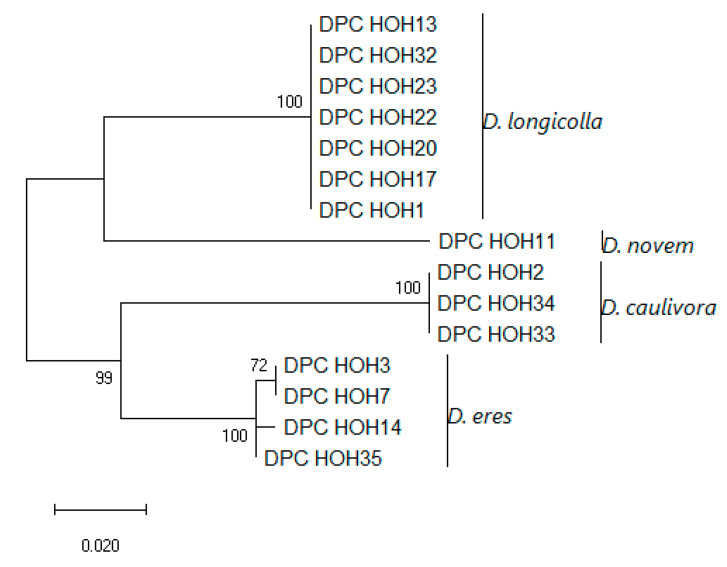
Evolutionary analysis of MAT1-2 DPC_HOH isolates by the Maximum Likelihood method based on *MAT1-2-1* sequences. The tree with the highest log likelihood (−583.18) is shown. The percentage of trees in which the associated taxa clustered together is shown next to the branches. The tree is drawn to scale, with branch lengths measured in the number of substitutions per site. This analysis involved 15 nucleotide sequences. There were a total of 266 positions in the final dataset.

**Figure 6 jof-11-00251-f006:**
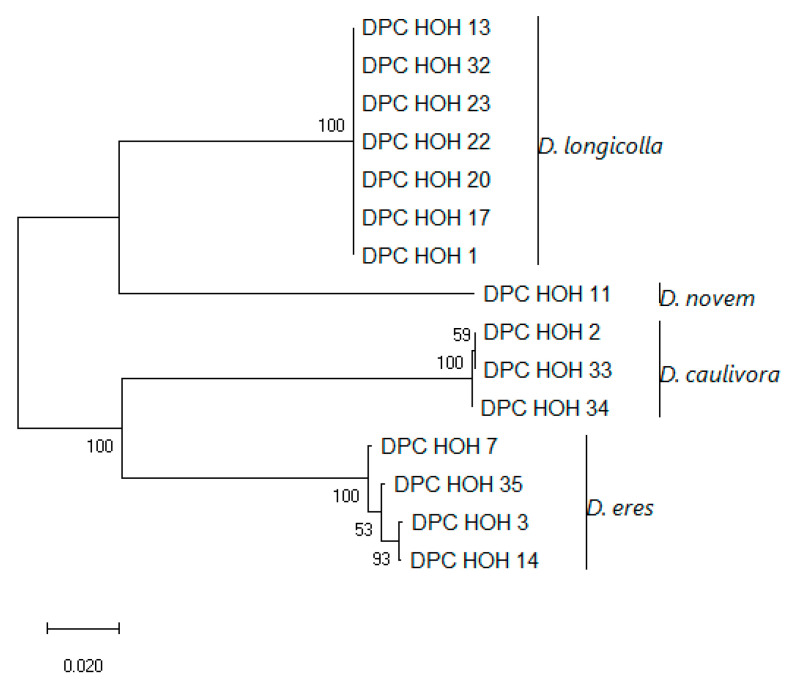
Evolutionary analysis of MAT1-2 DPC_HOH isolates by the Maximum Likelihood method based on ITS, *TEF*, *TUB*, and *MAT1-2-1* sequences. The tree with the highest log likelihood (−5245.28) is shown. The percentage of trees in which the associated taxa clustered together is shown next to the branches. The tree is drawn to scale, with branch lengths measured in the number of substitutions per site. This analysis involved 15 nucleotide sequences. There were a total of 1445 positions in the final dataset.

**Table 1 jof-11-00251-t001:** Primers used in this study ^1^.

Primer	Sequence 5′-3′	Use	Ref.
MAT1-1-1FW	GCA AMI GTK TIK ACT CAC A	Unspecific primers for amplification of *MAT1-1-1* in *Diaporthe* spp.	[4]
MAT1-1-1RV	GTC TMT GAC CAR GAC CAT		
MAT1-2-1FW	GCC CKC CYA AYC CAT TCA TC	Unspecific primers for amplification of *MAT1-2-1* in *Diaporthe* spp.	[4]
MAT1-2-1RV	TTG ACY TCA GAA GAC TTG CGT G		
DLmat111_fs	TCGAAGAGGAACGCAGGATC	Specific primers for amplification of *MAT1-1-1* in *D. longicolla*	This study
DLmat111_rs	GTCCAAGAGATCCACGGGGT		
DLmat121_fs	CAATCGCGTTTCTACGTCGG	Specific primers for amplification of *MAT1-2-1* in *D. longicolla*	This study
DLmat121_rs	TGTGTGTCCATCACTGCCTG		

^1^ Additional primers used for re-sequencing may be found in Appendix A: Primers for sequencing *D. caulivora* MAT1/2.

**Table 2 jof-11-00251-t002:** Accession numbers of MAT sequences used in phylogenies.

Isolate	Species	*MAT1-1-1* Acc. No.	*MAT1-2-1* Acc. No.
DPC_HOH1	*D. longicolla*	-	PP003228
DPC_HOH5		PP003219	-
DPC_HOH6		PP003220	-
DPC_HOH9		PP003221	-
DPC_HOH13		-	PP003230
DPC_HOH17		-	PP003231
DPC_HOH18		PP003222	-
DPC_HOH19		PP003223	
DPC_HOH20		-	PP003232
DPC_HOH22		-	PP003233
DPC_HOH23		-	PP003234
DPC_HOH26		PP003224	-
DPC_HOH29		PP003218	-
DPC_HOH32		-	PP003236
DPC_HOH2	*D. caulivora*	PP003215	PP003215
DPC_HOH33		PP003216	PP003240
DPC_HOH34		PP003217	PP003241
DPC_HOH3	*D. eres*	-	PP003237
DPC_HOH7		-	PP003238
DPC_HOH10		PP003225	-
DPC_HOH14		-	PP003239
DPC_HOH27		PP003227	-
DPC_HOH35		-	PP003235
DPC_HOH11	*D. novem*	-	PP003229
DPC_HOH16		PP003226	-

**Table 3 jof-11-00251-t003:** Isolates used in mating experiments.

Species	Isolate	Mating Type
*D. longicolla*	DPC_HOH17	MAT1-2
	DPC_HOH28	MAT1-1
	DPC_HOH29	MAT1-1
*D. eres*	DPC_HOH3	MAT1-2
	DPC_HOH7	MAT1-2
	DPC_HOH10	MAT1-1
*D. novem*	DPC_HOH11	MAT1-2
	DPC_HOH15	MAT1-1
	DPC_HOH16	MAT1-1

## Data Availability

Sequences underlying our analyses were submitted to GenBank; accession numbers are given in the main body of the manuscript. All other data are contained within the manuscript or in Appendix A.

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
