# Peer review of "Mating-Type Analysis in Diaporthe Isolates from Soybean in Central Europe"

_jof, 2025, doi:10.3390/jof11040251_

Round 1
Reviewer 1 Report (Previous Reviewer 2)
This manuscript is devoted to four Diaporthe species isolated from soybean in Central Europe. One of these species, D. eres, has a very wide plant host spectrum, often occurring also on forest trees. The research undertaken by the authors on mating types of these fungi is of great importance for mycology and plant pathology. It is a pity that the relationship between mating types and teleomorph production could not be explained. This indicates that there are still many problems to be explained in fungi. The manuscript is written carefully. The authors have taken into account most of the critical comments submitted to the first version of this manuscript. The manuscript should be published in JoF after a minor revision (see Remarks).
Remarks
Line 75 change [14] [5] to [14,5]
Line 129 ‘descriptions ……. are described’ - style correction recommended
Liner 17 ‘33 isolates of Diaporthe caulivora, D. eres, D. longicolla, and D. novem’, Lines 248-249 ‘After producing 32 isolates from Diaporthe spp. corresponding to D. caulivora, D. eres, D. longicolla, and D. novem’ – these data (32 vs. 33) should be compared and corrections should be made
Line 220 Colony PCR - this expression is unclear
Line 455 ‘…of >30 Diaporthe isolates’ – the exact number should be given
Line 593 J. Fungi, Line 625 J Fungi – the bibliographic data should be standardized (applies to all References)
Author Response
Major comments
This manuscript is devoted to four Diaporthe species isolated from soybean in Central Europe. One of these species, D. eres, has a very wide plant host spectrum, often occurring also on forest trees. The research undertaken by the authors on mating types of these fungi is of great importance for mycology and plant pathology. It is a pity that the relationship between mating types and teleomorph production could not be explained. This indicates that there are still many problems to be explained in fungi. The manuscript is written carefully. The authors have taken into account most of the critical comments submitted to the first version of this manuscript. The manuscript should be published in JoF after a minor revision (see Remarks).
Thank you very much for your kind comments. We are still waiting to see teleomorph structures. For D. longicolla we also talked to several researchers who heard or read about someone who saw sexual structures but could not replicate it. These “reports” fit with our findings but unfortunately are too vague to include in the publication.
Detail comments
Remarks
Line 75 change [14] [5] to [14,5]
Changed it to [5,14].
Line 129 ‘descriptions ……. are described’ - style correction recommended
Changed this. We now use the word “delineated”.
Liner 17 ‘33 isolates of Diaporthe caulivora, D. eres, D. longicolla, and D. novem’, Lines 248-249 ‘After producing 32 isolates from Diaporthe spp. corresponding to D. caulivora, D. eres, D. longicolla, and D. novem’ – these data (32 vs. 33) should be compared and corrections should be made
We agree that the occurrence of these two numbers is problematic. This appears like a contradiction but it is not. Indeed, we initially had 32 isolates and published about them in [9]. In this study we could only include 30 of them but added three more isolates. These additional isolates (DPC_HOH33, DPC_HOH34, and DPC_HOH35) are mentioned in section 2.1.
To fully resolve the apparent contradiction, we have added a sentence in line 253: “30 of the isolates described in [9] and the three additional isolates listed in section 2.1. were included in this study.”
Line 220 Colony PCR - this expression is unclear
In our opinion “colony PCR” is an established term in molecular biology. It describes the procedure where a small amount of bacteria from a colony on an agar plate, for example directly after a transformation, is transferred into a PCR reaction to act as template. No DNA or plasmid preparation is done in this case.
We added “(direct PCR on colonies of DH5α transformants)” to clarify the expression.
Line 455 ‘…of >30 Diaporthe isolates’ – the exact number should be given
We changed this to “33”. Thanks, we missed this when we changed it in the abstract.
Line 593 J. Fungi, Line 625 J Fungi – the bibliographic data should be standardized (applies to all References)
Thank you. We reviewed the references and found a few more inconsistencies. Seems we relied a bit too much on EndNote and maybe the editorial service.
Reviewer 2 Report (New Reviewer)
In recent publications on Diaprthe spp. (including species complexes such as D. eres, D. amygdali and D. arecaea), species boundaries are indicated using multi-gene phylogeny and coalescent-based species delimitation analyses (Hilário (2021a), Hilário (2021b) Pereira, D.S. et al. (2023), ). These concepts/methods are widely accepted in the definition of fungal species. On the other hand, in the biological species concept, mating, defined by mating-type genes, is essential for genome recombination and the generation of intraspecific diversity. Mating-type genes are known to be highly conserved within species, but differ between species. Furthermore, the important behaviours in mating of ascomycetes are homo- or hetero- sexual, which are determined by mating-type genes.
This paper describes the existence and structure of mating type genes in Diaphores isolates from soybean. As other reviewers have said, mating type genes of Diaporthe species have been analysed, but this paper shows new evidence that homothallic D. caulivora isolates consist of MAT1-1-1 and MAT1-2-1, although MAT1-1-1 could not be detected by PCR-based analysis in the previous study. In addition, D. longicolla has not been observed to have ascospores in the field, but all isolates tested had each mating type genes.
Unfortunately, the authors were unable to produce sexual reproductive bodies, but as they themselves point out, this does not necessarily mean that the fungus is incapable of sexual reproduction and cannot produce ascocarps or ascospores. It seems that the ability to produce conidia is important for the sexual response, so it may be a good idea to check the condition of the stock cultures they are using. In particular, the confirmation of sexual response in D. longicolla is a research topic that needs to be addressed in the future.
In conclusion, I think that the analyses of mating type genes in the genus Diaporthe presented in the manuscript are of value to readers.
On a minor point, when describing the coding DNA sequence, use the capitalised form CDS rather than cds. L340, L342.
Author Response
Major comments
In recent publications on Diaprthe spp. (including species complexes such as D. eres, D. amygdali and D. arecaea), species boundaries are indicated using multi-gene phylogeny and coalescent-based species delimitation analyses (Hilário (2021a), Hilário (2021b) Pereira, D.S. et al. (2023), ). These concepts/methods are widely accepted in the definition of fungal species. On the other hand, in the biological species concept, mating, defined by mating-type genes, is essential for genome recombination and the generation of intraspecific diversity. Mating-type genes are known to be highly conserved within species, but differ between species. Furthermore, the important behaviours in mating of ascomycetes are homo- or hetero- sexual, which are determined by mating-type genes.
This paper describes the existence and structure of mating type genes in Diaphores isolates from soybean. As other reviewers have said, mating type genes of Diaporthe species have been analysed, but this paper shows new evidence that homothallic D. caulivora isolates consist of MAT1-1-1 and MAT1-2-1, although MAT1-1-1 could not be detected by PCR-based analysis in the previous study. In addition, D. longicolla has not been observed to have ascospores in the field, but all isolates tested had each mating type genes.
Unfortunately, the authors were unable to produce sexual reproductive bodies, but as they themselves point out, this does not necessarily mean that the fungus is incapable of sexual reproduction and cannot produce ascocarps or ascospores. It seems that the ability to produce conidia is important for the sexual response, so it may be a good idea to check the condition of the stock cultures they are using. In particular, the confirmation of sexual response in D. longicolla is a research topic that needs to be addressed in the future.
Thank you very much for these suggestions. In time, we will do that. In the meantime, we are still waiting to see teleomorph structures. For D. longicolla we also talked to several researchers who heard or read about someone who saw sexual structures but could not replicate it. These “reports” fit with our findings but unfortunately are too vague to include in the publication.
In conclusion, I think that the analyses of mating type genes in the genus Diaporthe presented in the manuscript are of value to readers.
Thank you very much for your kind comments.
Detail comments
On a minor point, when describing the coding DNA sequence, use the capitalised form CDS rather than cds. L340, L342.
We changed this.
This manuscript is a resubmission of an earlier submission. The following is a list of the peer review reports and author responses from that submission.
Round 1
Reviewer 1 Report
Comments and Suggestions for Authors
This work provides a mating type analysis for four Diaporthe species from central Europe, and reveals that D. longicolla is heterothallic and D. caulivora has both mating type genes. The work might be publishable, but the current txt needs to be improved. Firstly, the taxonomy and the rules of species delimitation of Diaporthe have many changes in recent years, so the authors should check more latest references. Secondly, some parts of the results have been proved in other previous studies, and why do the authors repeat for it? Thirdly, the aims of the study are not very clear and difficult to show the importance which could strike on the readers. Since the primers didn’t work in amplifying some mating genes in this study, why not finding the answers in genomes and obtaining the organization of the mating-type loci for the species?
Comments on the Quality of English LanguageThe paragraphs should be more concise.
Reviewer 2 Report
Comments and Suggestions for Authors
This manuscript is devoted to four Diaporthe species isolated from soybean in Central Europe, some of which cause serious diseases of the plant and its seeds. The research undertaken by the authors on selected features (mating types) of these fungi is of great importance for science and, in longer term, for agricultural economy. These studies required a lot of time and work. The manuscript should be published in JoF. Some adjustments are needed beforehand as indicated in Remarks, for example. In general, the text is some places not very precise written, and there is even contradictory information therein. In "Results" certain methodological elements are provided, certain stages of the procedure are described, and this can be accepted because it improves the understanding of the text. However, fragments of text that should be in Discussion cannot be accepted in Results, this requires change.
Remarks
Title - is not fully clear and requires a fundamental change. The title suggests that a large number of Diaporthe species from Central Europe were studied, regardless of plant species. Do you research isolates only from soybean? The title in this case should be more general. For example, the results reported in Chapter 3.1 are not included at all in the current title of the paper
Line 14, line 59, line 64, line 67, spp. - in these cases it would be more correct to say Diaporthe species (several Diaporthe spp. = species plurum – it is double name of the same thing)
Line 16 more than 30 isolates – on page 6 there are 33 isolates listed (you can enter a specific number here)
Line 17 'several isolates', please provide a specific number.
Line 19 the species – do you mean this species?
Line 17-19 the authors sometimes contradict themselves, e.g. line 17-19' We found that different D. longicolla isolates either possess MAT1-1-1 or MAT1-2-1 making it heterothallic, which is in contrast to previous studies and the general assumption that the species only reproduces asexually' while in line 430 in Discussion you write 'All of this indicates that indeed D. longcolla may be purely anamorphic.'
Line 23 should summarize the results regarding D. eres and D. novem
Line 62 stains - it should be strains??
Line 72 ‘we showed’ – please note that the authors in the cited paper [8] are different than those in the current work. This form (we) cannot therefore be used.
Line 74 ‘Here we determined the mating types of these isolates’ – I understand that these are isolates representing four Diaporthe species – why are two Diaporthe species mentioned in the title – what about the rest? This text requires explanation and precise information
Line 80 ‘forms perithecia’ – in vitro? The information should be clarified
Line 93 'Because of this, it was only recently renamed from..' the first part of the sentence is illogical, the second part is correct
Line 110 it should be clarified what the purpose of research regarding D. eres and D. novem is
Line 119 spp. – please delete this, Diaporthe isolates will be correct
Line 158 – appropriate literature should be cited
Line 188 Table X ??
Line 196 Colony PCR - the text should be clarified
Line 221 provide the general title of chapter 3.1
Line 232-234 Compare this text, is it consistent with the text in lines 434-443 ?
Line 230-234, line 255-257, this text should be moved to Discussion
Line 247 '33 Diaporthe spp. isolates' earlier you write about 32 isolates (line 73)
Line 405, Line 410 – what is meant by 'heteromorph structures'?
Line 435 This sentence makes no sense and results from a misquote of the results of Santos et al 2011, who write "The PCR mating-type diagnosis showed that both mating-types exist in D. longicolla and both are present in the genome of a single isolate. However, no teleomorph was seen in pure cultures. The inability for isolates to form their sexual state together with the fact that the teleomorph of this species has never been found in nature indicates that this might be a purely anamorphic species.”
Line 435 you write that according to Santos et al. 2011 D. longicolla ‘must be purely anamorphic.’ While Santos et al. 2011 writes 'indicates that this might be a purely anamorphic species’. Please cite other authors correctly
Line 440 should be expressed precisely in the text because it causes ambiguity: in Line 434 you write an isolate of D. longicolla, in line 440 'in their isolates'. It is important whether Santos et al. [13] studied one isolate or many isolates
Line 445 "is probably determined elsewhere [1]' - please specify the text carefully and mention these reasons
Line 450 – what is the difference between your statement and the opinion of Santos et al. [13] – see line 435
Line 465 - please state whether you have intensively searched for soybean perithecia of D. longicolla in vivo, wouldn't this be a way to explain the described problem?
Line 502 this finding conforms with expectations - it's incomprehensible to me - was this the goal of your work?
Line 560 isles should be capitalized
Comments on the Quality of English Language
see Remarks
Reviewer 3 Report
Comments and Suggestions for Authors
The reviewed manuscript, titled "Mating type analysis in Diaporthe isolates from central Europe reveals that D. longicolla is heterothallic and that D. caulivora has both mating type genes” by Hosseini et al. is a relatively modest work that identifying the mating system in Diaporthe isolates. The title is quite accurate, however the work seems to have included a significant amount of analysis that do not seem to provide much worth to the readers. As an article I would reject this outright. I do believe that this could make a fine Communication style report – or another short form work – but this falls well short of the threshold for an Article in any journal.
Comments:
The introduction section is rather poorly structured. The authors have included a number of relevant facts about the Diaporthe spp. however the introduction reads primarily as a list of disparate facts. This should be overhauled to provide a clear, focused, cognizant structured section that will provide the readers with a clearer focus of the questions in the field and how the present work will address them. As structured this appears to separate each species bit-by-bit with little demonstration by the authors of how these can all interconnect. While nothing in this section is incorrect, it is very poorly organized. A typical introduction should begin broadly and then narrow down until the last paragraph explaining the aims and findings of the work. This whole section could just as easily be a simple table that compares each of the species and what is known and missing from each. As this is the case, a long section listing each one is not a substitute for a proper introduction.
The methods section does contain the necessary information to replicate experiments, however the authors refer to other publications and it would be necessary to do a significant amount of work for any researcher to completely replicate this work. Unless the authors can provide compelling evidence why they should not clearly articulate the steps and work throughout, I would require that the authors include their exact methodologies for all the work done herein. For example:
2.3.1: gross weight of mycelia for each preparation. Lysing protocol and conditions for mechanical disruption of cells. Give exact details of protocol followed that was adapted from ref 18. Provide details of the DNAseI digestion (units of enzyme and timing for incubations) and if any verification was performed to ensure complete DNA degradations.
2.3.2 provide recipe for your 25uL PCR reaction that you reference in 4. As you have changed the protocol, it would be better form to provide the details of what you had performed and reference 4 as what you adapted your work from. Provide primer concentration details for all reactions, not just a reference and then a change depending on what your reaction was amplifying.
2.3.3: elaborate on when the different elongation times were used. There is a rather large difference in the timings for this step (e.g. 15 vs. 45s can typically amplify about 2kB more with Phusion). Explain when and why the different times are used, presumably based on the different amplicon sizes. Figure 1 has relatively small fragments, so I am unsure why these different times were needed…. Etc.
Revise the rest of these sections for clarity and completion.
Figures 3-6 (the phylogenetic trees) do not seem to provide significant insight and I do not see any compelling reason why these would all be significant enough for inclusion in this work. I would move all to supplemental materials and perhaps include one tree that is based on all the limited data this work has generated.
The mating studies also do not seem compelling. The authors mentioned that they had trouble detecting the transcripts from the MAT locus, which typically become expressed to facilitate fusion and karyogamy, etc. As these were not expressed, it is not surprising that the authors did not seem to identify mating occurring. There are many conditions where fungi do not mate, of interest would be identification of conditions when they do. A much more compelling series of experiments would be to use genetics to overexpress these genes and demonstrate some functional outcomes as a result. There is a small nugget of work here that could lead to a very compelling work – but it needs significant development and testing to reach a publishable story.
Comments on the Quality of English LanguageThere are grammatical disagreements/tense issues, etc. that make the work a bit challenging to read in sections.